# Quasi-Solid-State SiO_2_ Electrolyte Prepared from Raw Fly Ash for Enhanced Solar Energy Conversion

**DOI:** 10.3390/ma15103576

**Published:** 2022-05-17

**Authors:** Gyo Hun Choi, Jaehyeong Park, Sungjun Bae, Jung Tae Park

**Affiliations:** 1Department of Chemical Engineering, Konkuk University, 120 Neungdong-ro, Gwangjin-gu, Seoul 05029, Korea; 2Department of Civil and Environmental Engineering, Konkuk Univesity, 120 Neungdong-ro, Gwangjin-gu, Seoul 05029, Korea; 3Department of Civil and Environmental Engineering, Institute of Construction and Environmental Engineering, Seoul National University, Gwanak-ro, Gwanak-gu, Seoul 08826, Korea

**Keywords:** raw fly ash, SiO_2_, quasi-solid-state electrolyte, solar energy conversion, dye-sensitized solar cells (DSSCs)

## Abstract

Quasi-solid-state electrolytes in dye-sensitized solar cells (DSSCs) prevent solvent leakage or evaporation and stability issues that conventional electrolytes cannot; however, there are no known reports that use such an electrolyte based on fly ash SiO_2_ (FA_SiO_2_) from raw fly ash (RFA) for solar energy conversion applications. Hence, in this study, quasi-solid-state electrolytes based on FA_SiO_2_ are prepared from RFA and poly(ethylene glycol) (PEG) for solar energy conversion. The structural, morphological, chemical, and electrochemical properties of the DSSCs using this electrolyte are characterized by X-ray diffraction (XRD), high-resolution field-emission scanning electron microscopy (HR-FESEM), X-ray fluorescence (XRF), diffuse reflectance spectroscopy, electrochemical impedance spectroscopy (EIS), and incident photon-to-electron conversion efficiency (IPCE) measurements. The DSSCs based on the quasi-solid-state electrolyte (SiO_2_) show a cell efficiency of 5.5%, which is higher than those of nanogel electrolytes (5.0%). The enhancement of the cell efficiency is primarily due to the increase in the open circuit voltage and fill factor caused by the reduced electron recombination and improved electron transfer properties. The findings confirm that the RFA-based quasi-solid-state (SiO_2_) electrolyte is an alternative to conventional liquid-state electrolytes, making this approach among the most promising strategies for use in low-cost solar energy conversion devices.

## 1. Introduction

In recent years, the recovery of SiO_2_ from industrial and agricultural waste (e.g., rice husk ash, semi-burned rice straw ash, bagasse ash, corn cob ash, coal fly ash, and bottom ash) has been extensively investigated due to the potential applications of SiO_2_ in chemical sensors and drug delivery, and as absorbents and catalyst supports [1,2,3,4,5]. Among the silica-enriched wastes, coal fly ash (FA) is generated at 1200–1700 °C during the combustion of coal in power plants and contains many metal oxides such as SiO_2_, Al_2_O_3_, Fe_2_O_3_, CaO, MgO, TiO_2_, Na_2_O, and K_2_O [6]. In order to address challenges associated with the large-scale production of FA (750 million tons per year) and its harmful impacts on the environment, a new reutilization strategy was developed to recover SiO_2_ from FA and use it in various applications. One study reported a combined organic acid/inorganic alkali/ultrasonication-assisted process for the extraction of SiO_2_ from FA; the prepared SiO_2_ showed good adsorption capacity toward methylene blue and malachite green [7]. Mesoporous silica (MCM-41) was prepared from FA using cetyltrimethylammonium bromide (CTAB) as a template; this FA-derived MCM-41, impregnated with polyethyleneimine, exhibited higher uptakes of CO_2_ than commercial silica products [8]. FA-derived mesoporous CdS/Al-MCM-41 composites were synthesized using the Si and Al extracted from FA, then applied as a photocatalyst for hydrogen production by the decomposition of H_2_O under visible light [9]. However, there is limited knowledge on the application of FA-derived SiO_2_ in solar energy conversion of renewable industrial applications.

Since the development of dye-sensitized solar cells (DSSCs) by the Grätzel group, DSSCs have received immense attention as a promising high-efficiency, low-cost energy source for solar energy conversion [10]. DSSCs comprise a transparent conductive substrate, a photoanode, sensitizers, electrolytes, and a counter electrode [11,12,13,14,15,16,17,18]. Electrolytes are a vital component of DSSCs, as they provide electrons that allow the sensitizers to return from an excited state to the ground state, thereby completing sensitizer regeneration. However, conventional liquid electrolytes in DSSCs present drawbacks that mostly stem from solvent leakage or evaporation issues and lower the stability of devices. Therefore, many studies have focused on replacing the conventional liquid electrolytes used in DSSCs with ionic liquids, hole-transport materials, and quasi-solid-state electrolytes in order to reduce the risk of leakage and environmental hazards [19,20,21]. Among them, the quasi-solid-state electrolytes based on low-molecular-weight polymers and inorganic nanomaterials exhibit ionic conductivities close to those of a conventional liquid electrolyte but maintain their quasi-solid-state electrolyte structures, thereby reducing solvent leakage and evaporation problems and enhancing mechanical properties. The excellent penetration of low-molecular-weight polymers into the photoanode is attributed to the improved interfacial contact between the photoanode/sensitizer surface and the quasi-solid-state electrolytes, which enables effective utilization of the sensitizer. Further, quasi-solid-state electrolytes offer advantages for electrochemical devices such as rechargeable batteries and supercapacitors in avoiding solvent leakage or evaporation and stability issues. Hence, it is expected that they can provide the same benefits when used with DSSCs as well.

In this work, FA_SiO_2_ was prepared from acetic acid-treated non-magnetic raw fly ash (RFA) and used as a quasi-solid-state electrolyte in DSSCs to improve their solar energy conversion efficiency. Detailed characterizations were conducted by X-ray diffraction (XRD), high-resolution field-emission scanning electron microscopy (HR-FESEM), X-ray fluorescence (XRF), diffuse reflectance spectroscopy, electrochemical impedance spectroscopy (EIS), and incident photon-to-electron conversion efficiency (IPCE) measurements. To the best of our knowledge, there has been no report on such a quasi-solid-state electrolyte based on FA_SiO_2_ from RFA for solar energy conversion applications.

## 2. Experimental

### 2.1. Materials and Chemicals

The raw fly ash (RFA) used in this investigation came from a South Korean thermal power plant. The pretreatment of RFA and the dissolution of silica and alumina sources present in the RFA were done with acetic acid (99.7%, DaeJung Chemicals & Metals Co., Ltd., Siheung-si, Korea) and NaOH (97.0 percent, DaeJung Chemicals & Metals Co., Ltd., Siheung-si, Korea). Amorphous silica was precipitated using hydrochloric acid (35.0 percent, DaeJung Chemicals & Metals Co., Ltd., Siheung-si, Korea). Deionized water (DIW, 18.3 M cm, Human Power I+, Human Corporation, Korea) was used in all studies. Aldrich Chemicals provided 1-Methyl-3-propylimidazolium iodide (MPII), lithium iodide, iodine, titanium diisopropoxide bis(acetylacetonate), H_2_PtCl_6_∙6H_2_O, poly(ethylene glycol) (PEG, Mn ~10,000 g/mol), and di di-tetrabutylammonium cis-bis(isothiocyanato)bis(2,2′-bipyridyl-4,4′-dicarboxylato)ruthenium(II) (N719). Dyesol provided the titania paste (18NR-T). Pilkington, France, provided fluorine-doped tin oxide (FTO) glass substrates (TEC8, 8/mm^2^, 3 mm thickness). J.T. Baker provided the solvents acetone, acetonitrile, ethanol, and isopropyl alcohol, as well as 1-butanol. Without additional purification, all chemicals and solvents were utilized as obtained.

### 2.2. Synthesis of FA_SiO_2_ from RFA

The overall synthetic procedures are illustrated in Figure 1. An exact amount of raw fly ash (RFA) (50.0 g) was added to a glass beaker containing 500 mL of DIW and mechanically stirred at 500 rpm for 1 h. After 1 h of water-washing, magnetic iron-rich fly ash (IRFA) particles were separated by placing a parafilm-wrapped neodymium magnet into the RFA-DIW suspension; the parafilm was detached from the magnet to collect the separated magnetic IRFA. This process was performed repeatedly until no more magnetic particles were attached to the parafilm-wrapped magnet. The separated IRFA and non-magnetic fly ash (NMFA) were washed with DIW three times by centrifugation (7000 rpm, 3 min) and oven-dried at 105 °C for 24 h. NMFA was pre-treated with acetic acid to remove minor impurities. An exact amount of NMFA (20.0 g) was mixed with 5 M acetic acid (200 mL) under continuous stirring (700 rpm) for 2 h at room temperature. After the pretreatment, the mixture of NMFA and acetic acid was centrifuged for solid-liquid separation. The supernatant was removed, and the acetic-acid-treated NMFA particles were washed with DIW three times and dried in an oven at 105 °C for 24 h. The acetic acid-treated NMFA (5.0 g) was mixed with 6.25 M of NaOH solution (mass ratio of 1:5) to extract silica, and this mixture was treated at 100 °C for 2 h, resulting in the formation of sodium silicate solution (SiO_2_ + 2NaOH → Na_2_SiO_3_ + H_2_O). After 2 h, this mixture was filtrated to attain a transparent filtrate of sodium silicate (water glass). Subsequently, 90.0 mL of the filtrate was mixed with 100.0 mL of DIW under magnetic stirring (1000 rpm) and preheated at 80 °C, which was constant for the duration of silica precipitation. Sodium silicate was then slowly neutralized with diluted HCl solution to pH 8 to precipitate silica (Na_2_O·xSiO_2_ + 2HCl → xSiO_2_↓ + 2NaCl + H_2_O). At this pH and a temperature of 80 °C, the mixture was additionally stirred for 2 h. The precipitated silica was collected and washed with DIW three times (7000 rpm, 3 min). Lastly, the solid was dried in an oven at 105 °C for 24 h and ground to obtain a white fly ash SiO_2_ (FA_SiO_2_) powder.

### 2.3. Preparation of Nanogel and Quasi-Solid-State (SiO_2_) Electrolytes

Particular attention was paid to 2 types of electrolytes for the fabrication of DSSCs: (1) a nanogel electrolyte consisting of PEG, MPII, LiI, and I_2_ in acetonitrile; (2) a quasi-solid-state electrolyte consisting of FA_SiO_2_, PEG, MPII, LiI, and I_2_ in acetonitrile. The mole ratio of ether oxygen (EO) to MPII, LiI was fixed at 20, the I_2_ content was fixed at 10 wt% with respect to the salt, and the FA_SiO_2_ content was fixed at 9 wt% with respect to PEG. (See Appendix A) In addition, for the DSSC based on 2 types of electrolytes, the electrolyte loading was about 0.05 mg cm^−2^.

### 2.4. Fabrication of DSSCs

The FTO glass was sonicated for 30 min before all fabrication methods with ethanol, acetone, and DIW. Spin-coating and the doctor-blade procedure were used to make a photoanode. Spin-coating the titanium diisopropoxide bis(acetylacetonate) solution (2 wt percent in 1-butanol) on the conducting side of the FTO glass at 1500 rpm for 20 s, followed by calcination at 450 °C for 30 min, resulted in a compact, thin TiO_2_ blocking layer on the FTO glass. The blocking layer was cast using commercial TiO_2_ paste using the doctor-blade process. A 3M sticky tape was used to adjust the film thickness. In order to prevent cracking during calcination, the substrate was warmed for 1 hour at 80 °C. A nanocrystalline TiO_2_ layer with a thickness of 7 μm was produced after calcination at 450 °C for 30 min. The TiO_2_ film layer was sensitized for 3 h in a dark environment using a 10^−4^ M ruthenium solution in ethanol. After that, the dye-sensitized photoanodes were submerged in 100% ethanol for 5 min to remove any remaining dye. Spin-coating an H_2_PtCl_6_ solution (1 wt percent in isopropyl alcohol) on the conducting side of the FTO glass at 1500 rpm for 20 s, followed by sintering at 450 °C for 30 min, yielded a platinum counter electrode. The photoanode and counter electrode were immediately cast with a quasi-solid-state electrolyte-based solution in acetonitrile for the DSSCs. Both electrodes were then overlaid and squeezed between two glass plates to allow the solvent and a thin electrolyte layer to evaporate slowly. According to previously documented techniques, the cells were placed in a vacuum oven for one day to allow the solvents to evaporate completely before being sealed with epoxy glue [22,23,24].

### 2.5. Characterization

HR-FESEM (SU8010, Hitachi High Technologies Corporation, Tokyo, Japan) with EDS was used to characterize the morphology of the produced FA SiO_2_ particles. XRD (JP/MAX-3C, Rigaku) was used to examine the mineral phases in RFA and FA SiO_2_ at 10–90° (2°/min). XRF (PANalytical, psilon3-XL) was used to examine the chemical components of RFA and FA SiO2. With a Brunauer–Emmett–Teller (BET) surface analyzer, the specific surface areas of the FA SiO_2_ samples were determined by nitrogen adsorption and desorption at −196 °C (3Flex, Micromeritics, Norcross, GA, USA). A spectrophotometer (Mega-900, Scinco) was used to produce UV–vis spectra in the range of 300–800 nm. Under one solar illumination, the photovoltaic properties of the cells were determined (100 mWcm^−2^, 150 W xenon lamp, measured from −0.1 V to 0.8 V). EIS measurements were taken with a CompactStat electrochemistry analyzer (Ivium Technologies) at frequencies ranging from 100,000 Hz to 0.1 Hz under one solar illumination and dark circumstances, respectively. A 150 W Xenon arc light source with a filter wheel and a 600 grooves/mm 500 nm blazed wavelength monochromator was used for the IPCE experiments (HS-Technologies). The photovoltaic results were also averaged over three samples.

## 3. Results and Discussion

### 3.1. Characterization of FA_SiO_2_

The XRD patterns of RFA and the synthesized FA_SiO_2_ are illustrated in Figure 1. The diffraction peaks of RFA are attributed to the occurrence of quartz (SiO_2_) and mullite (Al_6_Si_2_O_13_), and a broad hump (2θ = 20–35°) indicates the presence of amorphous aluminosilicate glass [25,26,27]. The result showed that RFA mainly consists of SiO_2_ and Al_2_O_3_, which is in good agreement with the XRF data for RFA in Table 1. The XRD pattern of the synthesized FA_SiO_2_ shows strong broad peaks centered at 2θ = 22–23°, indicating the characteristic amorphous nature of silica, which is consistent with the characteristics of amorphous silica produced from bentonite clay, rice straw ash, rice husk, and bagasse ash [28,29,30,31,32,33,34].

The morphology and particle size of the FA_SiO_2_ particles were observed by HR-FESEM at different magnifications, as illustrated in Figure 2. The images showed a non-crystalline structure of SiO_2_ (Figure 2a,b), which is in good agreement with the XRD pattern of FA_SiO_2_. The images also revealed that the FA_SiO_2_ nanoparticles are almost spherical in shape and aggregated to form bigger particles (Figure 2c,d), which is similar to the observations for typical amorphous silica produced from commercial sodium silicate solution [30]. EDS analysis was carried out, as shown in Appendix A. The elemental mapping images showed the location and quantities of the main elements (i.e., Si and O) and impurities contained in the extracted SiO_2_ (Appendix A). Dense and uniform distribution of Si and O was observed throughout all particles, indicating that SiO_2_ is the major constituent and relatively small amounts of impurities (i.e., Al, Fe, and Na) were mixed with SiO_2_. The EDS profile also revealed that the synthesized SiO_2_ comprised O (54.29%), Si (32. 02%), Al (7.55%), Na (5.53%), and Fe (0.61%). It is suggested that elements such as Al and Fe dissolve to form oxyanions under alkaline conditions during the extraction of Si source from acetic acid-treated NMFA [35]. The BET analysis showed that the specific surface area of the prepared SiO_2_ is 18.98 m^2^/g. The low surface area of the prepared FA_SiO_2_ in this study can be attributed to the fact that we did not use templates such as CTAB, which is commonly used for synthesizing mesoporous silica.

### 3.2. Characterization of Quasi-Solid-State (SiO_2_) Electrolyte

Figure 3a,b show the Nyquist and Bode phase plots of the DSSCs with the nanogel and quasi-solid-state (SiO_2_) electrolyte, respectively. By fitting the plots with Z-view software according to an equivalent circuit, as shown in Figure 3c, the corresponding electrochemical parameters including series resistance (R_S_), charge transfer resistance between the counterelectrode and electrolyte interface (R_ct1_), charge transfer resistance between dye-sensitized TiO_2_, the electrolyte (R _ct2_), the Warburg element (Z_w_), which indicates ionic diffusion in the electrolyte, minimum angular frequency (ω_min_), and a lifetime of electrons for recombination (τ_r_) are extracted and summarized in Table 2. The corresponding R_S_ values of DSSCs based on nanogel and quasi-solid-state (SiO_2_) electrolyte were 13.5 and 13.4, respectively, with no significant difference. On the other hand, a slight decrease in R_ct1_ and a remarkable decrease in R_ct2_ were observed in DSSCs based on quasi-solid-state (SiO_2_) electrolytes compared to the control group. These Nyquist results can be attributed to the increasement in the continuous free volumes related to redox ion mobility resulting from the interaction between the highly crystalline PEG polymer chain and FA_SiO_2_ nanoparticle [36]. Moreover, the middle frequencies in the Bode phase plot of nanogel and quasi-solid-state (SiO_2_) electrolytes shifted to lower frequencies; the corresponding ω_min_ values were 7.94 Hz for nanogel and 3.16 Hz for quasi-solid-state (SiO_2_) electrolytes. The τ_r_ values for DSSCs with nanogel and quasi-solid-state (SiO_2_) electrolytes were 20.0 and 50.4 ms, respectively. These results can be explained by the fact that the presence of FA_SiO_2_ nanoparticle in quasi-solid-state (SiO_2_) electrolyte-reduced electron recombination and improved electron transfer properties. By inserting the FA_SiO_2_ we obtained in the quasi-solid-state (SiO_2_) electrolyte, the FA_SiO_2_ nanoparticle can “hold” the electrolyte. This inorganic nanoparticle, which contains SiO_2_ and a small amount of Al_2_O_3_, weakly attracts Li^+^ ions to the surface due to their acidity. Further, the iodine ions accumulate around Li^+^ to attain electrical neutrality, resulting in the effect of forming a thin double layer [37]. It is assumed that these not only act as a charge transfer pathway, but also increase *Voc* by preventing the recombination of electrons in the TiO_2_ conduction band edge absorbed from the dye. Thus, the charge transfer resistance was reduced due to the role of the electron pathway, which is consistent with the increase in *FF*.

Figure 4a,b show the IPCE and diffuse reflectance spectra of DSSCs based on nanogel and quasi-solid-state (SiO_2_) electrolytes, respectively. In general, when an inorganic nanomaterial such as SiO_2_ is added to the electrolyte, the inorganic material acts to scatter incident light [38]. The same was expected in this study; however, the IPCE and reflectance spectra results did not indicate scattering effects. These results can be explained by the fact that the prepared FA_SiO_2_ nanoparticle was not large enough to cause scattering in DSSCs based on quasi-solid-state (SiO_2_) electrolytes. Moreover, this hypothesis is consistent with the nanoparticle size obtained from the previous HR-FESEM images.

### 3.3. Photovoltaic Performance of Quasi-Solid-State (SiO_2_) Electrolyte-Based DSSCs

The current density-voltage (*J-V*) characteristics were measured to investigate the effect of the quasi-solid-state (SiO_2_) electrolyte on the photovoltaic properties of DSSCs, as shown in Figure 5, and the results are summarized in Table 3. The *Jsc* (12.1 mA/cm^2^) values of the quasi-solid-state (SiO_2_) electrolyte-based cells were slightly lower than those of the nanogel electrolyte-based cell (12.6 mA/cm^2^). The reduced *Jsc* value of DSSCs based on quasi-solid-state (SiO_2_) electrolyte compared to the control group is presumably because the silica itself absorbs moisture from the atmosphere and is in direct contact with the dye on the TiO_2_ surface, causing some of the weakly attached dye to be desorbed [39]. We also observed that the *Voc* value of the quasi-solid-state (SiO_2_) electrolyte-based cells (0.73 V) was much larger than that of the nanogel electrolyte-based cell (0.65 V). One possible contribution to the improved *Voc* value includes the reduced charge recombination, electron back reaction loss by the FA_SiO_2_ in quasi-solid-state (SiO_2_) electrolyte. In addition, the onset of the dark current shifted to a higher forward bias in the presence of the quasi-solid-state (SiO_2_) electrolyte, compared to the control group, as shown in Appendix A. This supports well that the charge recombination reaction between the photoanode and the redox ions in the quasi-solid-state (SiO_2_) electrolyte is effectively suppressed, which gives rise to an enhancement in the *Voc* value. In addition, when the electrolyte was prepared with a FA_SiO_2_ as the quasi-solid-state materials, the *FF* value was further enhanced from 0.60 to 0.62. The increase in *FF* is due to the electron-enhanced electron pathway in DSSCs based on quasi-solid-state (SiO_2_) electrolyte, which results in reduced charge transfer resistance [40]. Upon using FA_SiO_2_ as a quasi-solid-state (SiO_2_) electrolyte, a high energy conversion efficiency of 5.5% at 100 mW/cm^2^ could be achieved. Consequently, quasi-solid-state (SiO_2_) nanoparticles in electrolyte compensated for reduced *Jsc* value due to their not only enhanced electron pathway but also minimized interfacial charge recombination loss. For reliability, we provided the photocurrent density-photovoltaic curves of DSSCs of the different sets of quasi-solid-state (SiO_2_) electrolytes (Appendix A). In addition, the photovoltaic efficiency of the DSSCs based on FA_SiO_2_ as a quasi-solid-state (SiO_2_) electrolyte represents one of the highest values reported for quasi-solid-state electrolyte-based ssDSSCs to date, as shown in Appendix A [41,42,43,44,45]. The photovoltaic properties of DSSCs of FA_SiO_2_ with different contents are shown in Appendix A, and the results are summarized in Appendix A. The quasi-solid-state (SiO_2_) 3 and 6 wt% electrolytes-based DSSCs have low redox ion mobility properties due to the small amount of FA_SiO_2_ that does not allow the efficiently enhanced free volumes. Meanwhile, higher FA_SiO_2_ concentration (12 wt%) in the quasi-solid-state electrolyte significantly decreases solar energy conversion efficiency due to complete solidification of the electrolyte. Moreover, optimizing our suggested quasi-solid-state electrolyte-based DSSC with functionalized scattering layers as a light-harvesting structure results in an increase in the beam path of solar irradiation within the photoanode due to scattering in the device, which we believe improves the light-harvesting efficiency of the DSSCs [46]. Therefore, the RFA-based quasi-solid-state (SiO_2_) electrolyte was able to improve the solar energy conversion properties based on the reduced electron recombination and enhanced electron transfer properties.

## 4. Conclusions

This study presents a novel method for the preparation of FA_SiO_2_ from RFA for use as a quasi-solid-state electrolyte in DSSCs. The condition of the acetic acid-treated NMFA used in the synthesis controls the morphology and crystal structure of the amorphous SiO_2_, leading to significant differences in their physical, chemical, and electrochemical properties. Compared to the nanogel electrolyte, the quasi-solid-state (SiO_2_) electrolyte features reduced electron recombination and enhanced electron transfer properties. Accordingly, high solar energy conversion efficiencies of 5.5% were achieved using this electrolyte, corresponding to enhancements of 10% compared to nanogel electrolyte-based DSSCs. The RFA-based quasi-solid-state (SiO_2_) electrolyte can be an alternative to conventional liquid-state electrolytes, and the approach represents one of the most promising strategies for its use in low-cost solar energy conversion devices. Our experimental results suggest that energy-related solid waste, such as RFA, can be converted into energy-production materials. Additionally, this application for DSSCs fabrication can provide a novel insight that RFA-based Si products can be used to develop cost-effective and innovative active layers for Si solar cells, perovskite solar cells, and organic solar cells with improved photovoltaic performance.

## Data Availability

No data available.

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
