# Peer review of "Quasi-Solid-State SiO2 Electrolyte Prepared from Raw Fly Ash for Enhanced Solar Energy Conversion"

_materials, 2022, doi:10.3390/ma15103576_

Round 1

Reviewer 1 Report

The manuscript from Choi et al shown a cell efficiency of 5.5 % by using a quasi solid state electrolyte (SiO2). This is not in the state of art of the technology (https://doi.org/10.1016/j.jiec.2021.12.009)

Please state why you are not reaching over 8% efficiency and why instead you are getting 5.5%.

Author Response

Thank you very much for your careful review on our manuscript. We revised the manuscript mostly as suggested.

Reviewer 2 Report

In this study, the authors used fly-ash SiO2 to prepare quasi-solid state DSSCs. The quasi-solid state cells have higher power conversion efficiency than those without fly-ash SiO2. Some arguments mentioned in the study need to be further verified because of insufficient analyses. The following points are helpful for the authors to improve the manuscript:

  1. In order to verify the effect of FA-SiO2, the cells with various FA-SiO2 contents should be shown to display asymptotic efficiency, not only 0 and 9 wt% cases.
  2. The experimental information is insufficient. For example,
  • How much PEG and acetonitrile were used in the electrolyte?
  • What is the ratio of MPII to LiI?
  • In the last line of page 6, what is "the salt"? (10 wt% respect to the salt)
  • How did the authors control the thickness of the electrolyte layer?
  1. Because the FA-SiO2 is not a pure chemical, deviations in composition and performance should be provided.
  2. The explanation for the semicircles in the high- and low-frequency region in the Nyquist plot is different from that reported in the literature. Why will the change in the electrolyte influence both semicircles? Did the authors re-check it?
  3. The FA-SiO2 cell has worse IPCE but better efficiency. The result needs further explanation.
  4. Some typos. For example, the reference number should be put in front of the comma and period.

Author Response

(The authors gave the same response as above.)

Reviewer 3 Report

The subject of the paper is the quasi solid-state electrolytes based on FA_SiO2 are prepared from RFA and poly(ethylene glycol) (PEG) for solar energy conversion. The raw fly ash (RFA) used in this study was obtained from a thermal power plant in South Korea, and the DSSCs based on the quasi solid-state electrolyte (SiO2) show a cell efficiency of 5.5 %. However, some recommendations should be taken into account for publication:

Originality/Novelty

While the fly ash material is obviously unique to the study and the numbers of papers in the area of DSSCs are few, the methods used to fabricate the DSSCs are not particularly original or novel.

Quality of Presentation

The overall presentation quality is acceptable. The article is, on the whole, well-written; the English used in the paper is understandable.

  • Methodology

Nothing is said about the reproducibility of the measurements. How many samples were measured during this study?

  • Results and Discussion

“The reduced Jsc value of DSSCs based on quasi solid-state (SiO2) electrolyte compared to the control group is presumably because the silica itself absorbs moisture from the atmosphere and is in direct contact with the dye on the TiO2 surface, causing some of the weakly attached dye to be desorbed.” is sufficiently backed up with a reference. This should be rectified in future versions of the paper.

It is best to compare your study findings with those of the existing literature and then to provide two to three examples from other studies that demonstrate what is similar to your study findings and what is different, particularly with regard to conventional electrolytes, on every figure and table related to your study.

Conclusions

The conclusions are straightforward. This needs to be improved. Include more specifics about the outcomes that were achieved.

Overall Merit:

The article certainly has some merit. For the rest, I believe that the article is organised in a logical and understandable manner.

Author Response

Thank you very much for your careful review on our manuscript. We revised the manuscript mostly as suggested:

Round 2

Reviewer 1 Report

Authors did not adress my revision. 

Author Response

Thank you very much for your careful review on our manuscript. We revised the manuscript mostly as suggested:

Reviewer # 1

< Comments >

Authors did not address my revision.

< Response >

As the reviewer pointed out, there is a report on high-performance quasi solid-state electrolyte-based DSSCs, which include nanofiller such as SiO2. However, in the reference, the author used a high thickness (9.2 ± 0.4 µm) photoanode structure, and they added a functional light-harvesting structure on conventional photoanode as a scattering layer which helps to improve the light reflection property. On the other hand, we want to highlight the importance of fly ash SiO2 (FA_SiO2) nanofiller from raw fly ash (RFA). Possibly, our suggested quasi solid-state electrolyte-based DSSCs could further improve the photovoltaic performance if added to the conventional TiO2 scattering layers. So, we would like to cite this given reference as a method for improving photovoltaic performance for DSSCs.

To address this comment, we added the following descriptions and cited the paper as Ref. #46.

“Also, optimizing our suggested quasi solid-state electrolyte-based DSSC with functionalized scattering layers as a light-harvesting structure results in an increase in the beam path of solar irradiation within the photoanode due to scattering in the device, which we believe improves the light-harvesting efficiency of the DSSCs [46].”

[46] Venkatesan, S.; Chen, Y.-Y.; Chien, C.-Y.; Tsai, M.-H.; Teng, H.; Lee, Y.-L. Composite electrolyte pastes for preparing sub-module dye sensitized solar cells, J. Ind. Eng. Chem. 2022, 107, 383.

Reviewer 2 Report

The authors have responded to the questions and revised most issues. However, there are still some minor points:

  1. According to Table S3, the curves for 3 wt% and 12 wt% are wrong, it needs to change each other.
  2. 36 is inappropriate. The authors shouldn’t use their previous paper to explain current doubts. Please use others’ papers to support your viewpoints.

Author Response

Thank you very much for your careful review on our manuscript. We revised the manuscript mostly as suggested:

Reviewer # 2

<General Comments>

The authors have responded to the questions and revised most issues. However, there are still some minor points:

<Specific Comments>

(1) According to Table S3, the curves for 3 wt% and 12 wt% are wrong, it needs to change each other.

< Response >

To address this comment, we modified the Table S3.

Table S3. Photovoltaic properties of DSSCs based on quasi solid state (SiO2) electrolytes with different FA_SiO2 contents under 1 sun illumination. (AM 1.5 G, 100 mW/cm2).

Voc (V)

Jsc (mA/cm2)

FF

h (%)

Quasi solid state (SiO2) 3wt%

0.67

11.6

0.65

5.0

Quasi solid state (SiO2) 6wt%

0.64

10.9

0.60

4.2

Quasi solid state (SiO2) 9wt%

0.73

12.1

0.62

5.5

Quasi solid state (SiO2) 12wt%

0.62

10.0

0.60

3.7

(2) 36 is inappropriate. The authors shouldn’t use their previous paper to explain current doubts. Please use others’ papers to support your viewpoints.

< Response >

To address this comment, we added the following descriptions and cited the paper as Ref. #36.

“This Nyquist results can be attributed to the increasement in the continuous free volumes related to redox ion mobility resulting from the interaction between the highly crystalline PEG polymer chain and FA_SiO2 nanoparticle [36].”

[36] Dong, R.-X.; Shen, S.-Y.; Chen, H.-W.; Wang, C.-C.; Shih, P.-T.; Liu, C.-T.; Vittal, R.; Lin, J.-J.; Ho, K.-C. A novel polymer gel electrolyte for highly efficient dye-sensitized solar cells, J. Mater. Chem. A 2013, 1, 8471.

Reviewer 3 Report

The authors have addressed all of the concerns that were raised.

Author Response

Thank you very much for your careful review on our manuscript. We revised the manuscript mostly as suggested:

Reviewer # 3

< Comments>

The authors have addressed all of the concerns that were raised.

< Response >

We truly appreciate the reviewer’s comment.
